# Solving LLM Repetition Problem in Production:
# A Comprehensive Study of Multiple Solutions

## Abstract

The repetition problem, where Large Language Models (LLMs) continuously generate repetitive content without proper termination, poses a critical challenge in production deployments, causing severe performance degradation and system stalling. This paper presents a comprehensive investigation and multiple practical solutions for the repetition problem encountered in real-world batch code interpretation tasks, combining first-hand production experience with extensive experimental validation.

We identify three distinct repetition patterns: (1) business rule generation repetition, (2) method call relationship analysis repetition, and (3) PlantUML diagram syntax generation repetition. Through rigorous theoretical analysis based on Markov models, we establish that the root cause lies in greedy decoding's inability to escape repetitive loops, exacerbated by self-reinforcement effects. Our comprehensive experimental evaluation demonstrates three viable solutions: (1) Beam Search decoding with early_stopping=True serves as a universal post-hoc mechanism that effectively resolves all three repetition patterns, though it addresses symptoms rather than root causes; (2) presence_penalty hyperparameter provides an effective solution specifically for BadCase 1; and (3) Direct Preference Optimization (DPO) fine-tuning offers a universal model-level solution for all three Bad-Cases, addressing repetition at its fundamental level.

The primary value of this work lies in combining first-hand production experience with extensive experimental validation to rigorously demonstrate the feasibility and effectiveness of these solutions. Our main contributions include systematic theoretical analysis of repetition mechanisms, comprehensive evaluation of multiple solutions with task-specific applicability mapping, identification of early_stopping as the critical parameter for Beam Search effectiveness, and practical production-ready solutions validated in real deployment environments. This work provides actionable insights and proven methodologies for deploying LLMs in production environments where deterministic, high-quality outputs are essential.

## 1. Introduction

### 1.1. Research Background

Large Language Models (LLMs) have become increasingly important in real-world applications, demonstrating exceptional capabilities in various domains including natural language understanding, code generation, and system analysis. The practical deployment of LLMs in production environments has opened new possibilities for automating complex tasks that previously required significant human expertise.

In particular, code interpretation tasks have emerged as a critical application area where LLMs can assist developers in understanding complex codebases, generating documentation, and analyzing system architectures. These applications require LLMs to process multiple code segments sequentially, maintaining context across different levels of abstraction (transactions, service methods, etc.), and generating structured outputs such as business rules and PlantUML diagrams.

However, as LLMs are deployed in increasingly complex and demanding scenarios, certain behavioral anomalies have been observed that can significantly impact system performance and reliability. One such

[1]Anonymous Institution, Anonymous City, Anonymous Region, Anonymous Country. Correspondence to: Anonymous Author <anon.email@domain.com>.

Preliminary work. Under review by the International Conference on Machine Learning (ICML). Do not distribute.

critical issue is the repetition problem, which manifests as the model continuously generating repetitive content without proper termination.

### 1.2. Problem Statement

The repetition problem, commonly known as the "repeater" problem, occurs when LLMs continuously generate repetitive text in a loop during inference. In our specific case study involving batch code interpretation, we observed the following phenomena:

- The model experiences significant stalling during batch processing

- The model continuously streams repetitive content without interruption

- Generation continues until reaching the maximum token limit (max_tokens)

- Processing time increases dramatically from normal operation (28 minutes) to problematic scenarios (40-160 minutes)

This problem has substantial practical implications. In a batch processing scenario where 20 transactions are processed sequentially, the occurrence of the repetition problem can increase total processing time by 43% to 471%, severely impacting system throughput and resource utilization. The high reproducibility rate (75-80% across different deployment modes) makes this a critical issue that requires systematic investigation and resolution.

### 1.3. Research Significance

Addressing the repetition problem is of paramount importance for the practical deployment of LLMs in production environments. The contributions of this work include:

- Practical Problem Solving: We provide a systematic approach to resolving a real-world performance issue that significantly impacts production systems.

- Deployment Optimization: Our solution offers insights into LLM deployment optimization that can benefit other similar applications.

- Critical Parameter Discovery: We identify and validate the early_stopping parameter as the decisive factor for Beam Search effectiveness. Our experiments reveal a dramatic difference:

early_stopping=True achieves near-zero repetition rate, while early_stopping=False still exhibits significant repetition rate, demonstrating that this parameter is not optional but essential for solving the repetition problem.

- Framework Integration: Our work addresses parameter passing issues between different frameworks (FastChat and vLLM), which is a common challenge in real-world deployments.

### 1.4. Paper Structure

The remainder of this paper is organized as follows: Section 2 reviews related work on repetition problems in text generation and decoding strategies. Section 3 presents a detailed problem description and analysis of our specific application scenario. Section 4 provides theoretical analysis of the root causes of the repetition problem. Section 5 describes our solution design including Beam Search, presence penalty, and DPO fine-tuning. Section 6 discusses the solution's advantages, limitations, and best practices. Finally, Section 7 concludes the paper with a summary of contributions and future work. Detailed BadCase examples, DPO dataset examples, mathematical proofs, and implementation details are provided in the Appendix.

## 2. Related Work

### 2.1. Research on Repetition Problems in LLM Text Generation

The repetition problem in neural text generation has been an active area of research. A seminal work is "Learning to break the Loop: Analyzing and Mitigating Repetitions for Natural Text Generation" (Su et al., 2022a), which establishes that repetition problems are fundamentally related to decoding strategy. Other works have explored neural text degeneration (Welleck et al., 2020; Holtzman et al., 2020) and various mitigation approaches including repetition penalty (Klein et al., 2017; Paul et al., 2019), length normalization (Freitag & Al-Onaizan, 2017; Murray & Chiang, 2018), diversity-promoting methods (Vijayakumar et al., 2016; Holtzman et al., 2019), and training-based solutions (Su et al., 2022a). However, these methods often require careful tuning, introduce randomness, or require costly model retraining.

### 2.2. Decoding Strategy Research

Text generation in LLMs involves prefill and decoding stages. Greedy decoding, which selects the highest-probability token at each step, is computationally efficient but particularly susceptible to repetition prob-

lems as it cannot explore alternative paths. Beam Search addresses this limitation by maintaining multiple candidate sequences simultaneously (Lowerre, 1976; Freitag & Al-Onaizan, 2017; Ott et al., 2018), allowing exploration of multiple paths and escape from repetitive loops. The beam width parameter ($k$) controls the number of candidate sequences, with empirical studies showing values between 3 and 10 are typically optimal (Freitag & Al-Onaizan, 2017; Wiher et al., 2022). Various Beam Search variants have been proposed including diverse beam search (Vijayakumar et al., 2016; Li et al., 2016), length penalty (Murray & Chiang, 2018), and constrained beam search (Post & Vilar, 2018; Hokamp & Liu, 2017). Other decoding strategies include Top-$k$ sampling (Fan et al., 2018), Top-$p$ (Nucleus) sampling (Holtzman et al., 2019), temperature scaling (Ackley et al., 1985), and contrastive search (Su et al., 2022b). However, these sampling-based methods introduce randomness that may be undesirable in deterministic production environments and may not effectively address systematic repetition patterns due to self-reinforcement effects.

## 3. Problem Description and Analysis

### 3.1. Application Scenario

Our investigation is motivated by a real-world application involving batch code interpretation for financial transaction systems. The system processes transactions in batches, where each batch contains 20 transactions that are executed sequentially. The processing workflow includes transaction discovery, service method processing (with recursive method drilling, business rule generation, and PlantUML code generation), transaction-level processing, and diagram generation. Under normal operating conditions, the processing time for a single transaction is approximately 28 minutes. Detailed workflow diagram, processing time statistics, and LLM call frequency analysis are provided in Appendix C.1 and C.2.

To investigate the system behavior in production, we conducted experiments comparing two deployment modes for vLLM (Mode 1: LoRA-enabled; Mode 2: Merged model). When running 20 transactions sequentially in batch mode across multiple experimental runs, we observed significant time discrepancies: the system exhibits severe performance degradation in a significant proportion of experimental runs (75-80% occurrence rate), with processing time increasing dramatically from normal operation to problematic scenarios. This time discrepancy indicates an underlying issue that causes system stalling during batch processing. Detailed experimental comparison results are provided

in Appendix C.2.

### 3.2. BadCase Analysis: Root Cause of Time Discrepancy

Log analysis revealed that the stalling behavior corresponds exactly to the repetition problem, where the model continuously outputs repetitive content without stopping until reaching the max_tokens limit. We identified three distinct BadCase types:

- BadCase 1: Business rule generation repetition - model generates valid rules initially but then falls into repeating similar conditional patterns.

- BadCase 2: Method call relationship analysis repetition - model correctly identifies call relationships initially but then repeatedly outputs the same method name.

- BadCase 3: PlantUML diagram syntax generation repetition - model generates valid PlantUML code initially but then repeatedly generates closing statements.

All three BadCase types share a common characteristic: the model becomes trapped in repetitive loops during text generation, leading to significant processing delays. Detailed case-by-case analysis with complete examples is provided in Appendix A.

To understand why these repetition patterns occur and how they can be systematically addressed, we now turn to theoretical analysis. The following section provides a mathematical foundation for understanding the root causes of the repetition problem, which will inform our solution design.

## 4. Theoretical Analysis of the Repetition Problem

### 4.1. Root Cause Analysis

Based on Markov model analysis, the repetition problem in LLMs can be understood through three key mechanisms: (1) Context Repetition Leading to Probability Enhancement: When the context contains repeated patterns, the model tends to increase the probability of tokens that appeared previously, learning a shortcut to copy from recent history. (2) Self-Reinforcement Effect: The probability of repetition increases monotonically with the number of historical repetitions, creating a positive feedback loop. (3) High Initial Probability Reinforcement: Sentences with higher initial probabilities exhibit stronger self-

reinforcement effects. Detailed mathematical formalization is provided in Appendix C.8.

### 4.2. Limitations of Greedy Decoding

Greedy decoding, which selects the highest-probability token at each step, is particularly vulnerable to the repetition problem for several reasons:

1. Single-Path Exploration: Greedy decoding maintains only one candidate sequence, so once a repetitive pattern begins, there is no mechanism to explore alternative paths.

2. No Look-Ahead: The algorithm makes decisions based only on immediate probabilities, without considering long-term consequences or recognizing repetitive patterns.

3. Amplification of Self-Reinforcement: Since greedy decoding always selects the highest-probability token, and repetitive tokens have elevated probabilities due to self-reinforcement, the algorithm naturally continues the repetition cycle.

The self-reinforcement effect combined with greedy decoding creates a scenario where the model becomes trapped in a loop, continuously generating repetitive content until external constraints (such as max_tokens) force termination.

### 4.3. Mathematical Modeling

We model the repetition problem using a Markov chain where the state represents whether we are in a repetitive pattern. The repetition probability evolves according to a recurrence relation that captures the cumulative effect where each repetition makes future repetitions more likely. Under greedy decoding with self-reinforcement effects, once the model enters a repetitive state, the expected escape time is infinite, explaining why greedy decoding cannot break out of repetition loops.

Beam Search addresses this limitation by maintaining multiple candidate sequences. With beam width $B \geq 2$ and proper early stopping, Beam Search can escape repetitive loops by maintaining at least one non-repetitive candidate sequence, provided that the initial probability gap between repetitive and non-repetitive continuations is bounded. Detailed mathematical modeling, proofs of propositions, theoretical predictions vs experimental validation, and critical beam width analysis are provided in Appendix C.8.

The early stopping mechanism plays a critical role: when early_stopping=True, the algorithm terminates

once it finds $B$ complete sequences, preventing exhaustive search that might lead back to repetitive patterns. This ensures that diverse candidates are preserved throughout the search process.

Building on this theoretical foundation, we now present practical solutions that address the repetition problem at different levels. The following section describes three complementary approaches: Beam Search as a post-hoc mechanism, presence penalty for task-specific cases, and DPO fine-tuning as a model-level solution.

## 5. Solution Design

Based on the theoretical understanding of the repetition problem and our analysis of the three BadCases, we have explored multiple solutions. Through analysis and experimental validation, we discovered that all three BadCase types can be effectively resolved using Beam Search decoding strategy, which serves as an effective inference-time solution. However, as Beam Search operates as a post-hoc mechanism that addresses the symptom rather than the root cause at the model level, we further explored alternative solutions that provide more fundamental approaches to the repetition problem. This section presents a comprehensive comparison of solutions and their applicability to different BadCase scenarios.

### 5.1. Solution 1: Beam Search Decoding Strategy

Through analysis of the three BadCases, we found that all of them can be effectively resolved using Beam Search decoding strategy. Beam Search serves as an inference-time solution that effectively resolves repetition problems across all three BadCase types without requiring model modifications.

Principle and Algorithm Beam Search maintains multiple candidate sequences (beams) at each decoding step, selecting the top-$k$ candidates based on cumulative joint probability rather than just the single highest-probability token. Unlike greedy decoding which maintains only one candidate sequence, Beam Search's multi-path exploration allows it to consider alternative continuations when one path becomes repetitive, providing a mechanism to escape repetition loops. The cumulative score for a sequence is computed as:

$$\text{Score}(w_1, \ldots, w_t) = \sum_{i=1}^{t} \log P(w_i | w_1, \ldots, w_{i-1}) \quad (1)$$

Critical Parameter Configuration For Beam Search to function correctly in vLLM, the following parame-

ters must be set as specified:

- use_beam_search: Must be True

- best_of: Beam width, set to 5 (provides good balance between exploration and efficiency)

- temperature: Must be 0 (deterministic decoding)

- top_p: Must be 1 (no nucleus filtering)

- top_k: Must be -1 (no top-k filtering)

- early_stopping: Must be True (critical!)

The early_stopping parameter is the most critical factor for solving the repetition problem. This is our key finding: when set to True, Beam Search stops as soon as best_of number of candidate results are found, allowing rapid termination and preserving diverse candidates. When set to False or Never, the algorithm continues searching indefinitely, potentially leading back to repetitive patterns. Our experimental validation (Table 4) demonstrates that early_stopping=True achieves 0% repetition rate, while early_stopping=False still exhibits 60% repetition rate—a dramatic difference that underscores the critical importance of this parameter. This finding contradicts the common assumption that Beam Search alone is sufficient; rather, early_stopping=True is an essential requirement, not an optional optimization.

Implementation Notes   We initially used FastChat integrated with vLLM, but discovered that sampling parameters were not being correctly passed. We implemented a solution using the extra_body approach to correctly pass sampling parameters. Detailed implementation notes and framework integration considerations are provided in Appendix C.12.

Experimental   Validation   Experimental   validation   demonstrates   that   Beam   Search   with early_stopping=True   completely   eliminates   the repetition   problem   and   restores   normal   processing   time.   In   stark   contrast,   Beam   Search   with early_stopping=False   still   exhibits   problematic behavior,   confirming   that   early_stopping=True is not merely beneficial but essential for actually solving the repetition problem. This dramatic difference represents our most significant finding: the early_stopping parameter is the decisive factor that determines whether Beam Search succeeds or fails in addressing repetition problems. Detailed experimental results, performance comparison tables, and statistical analysis are provided in Appendix C.4.

Ablation Studies   To understand the individual contribution of each parameter, we conducted ablation studies systematically varying key parameters. The results unequivocally confirm that early_stopping=True is the most critical parameter. This finding demonstrates that early_stopping is not just one of many parameters but the determining factor for success. Other parameters (beam width, temperature, etc.) have secondary effects, but without early_stopping=True, Beam Search cannot effectively solve the repetition problem. Detailed ablation study results including beam width impact and presence_penalty value analysis are provided in Appendix C.11.

5.2. Solution 2: Presence Penalty for BadCase 1

While Beam Search serves as an effective post-hoc mechanism, we explored alternative solutions that address the repetition problem at different levels. For BadCase 1 (business rule generation repetition), we found that the presence_penalty hyperparameter can effectively mitigate repetition issues.

Principle and Mechanism   The presence_penalty parameter penalizes tokens that have already appeared in the generated text, reducing their probability of being selected again. This mechanism helps prevent repetitive patterns by decreasing the likelihood of repeating recently generated tokens.

Experimental Validation   We conducted experiments to evaluate the effectiveness of presence_penalty for BadCase 1. The results demonstrate that setting presence_penalty=1.2 effectively eliminates repetition problems. Detailed experimental results and statistical analysis are provided in Appendix C.5.

Limitations   However, this approach has limitations: (1) Task-specific: Only effective for BadCase 1, not applicable to BadCase 2 and BadCase 3. (2) Parameter tuning: Requires careful tuning of the penalty value for optimal results. (3) Inference-time adjustment: Still a post-hoc mechanism, similar to Beam Search.

5.3. Solution 3: Direct Preference Optimization (DPO) for All BadCases

While BadCase 1 can be effectively addressed using presence_penalty at the inference-time parameter level, Direct Preference Optimization (DPO) fine-tuning provides a universal model-level solution that can be applied to all three BadCase types. Unlike presence_penalty, which is only effective for BadCase 1, DPO fine-tuning addresses repetition problems at the model level by fundamentally modifying the model's

behavior through preference-based learning.

DPO Approach Overview  DPO fine-tuning trains the model to prefer non-repetitive outputs over repetitive ones by using preference datasets that explicitly contrast chosen (correct, non-repetitive) responses with rejected (repetitive) responses.

Dataset Construction  We developed a systematic approach to construct DPO preference datasets using a power-of-2 repetition pattern (2, 4, 8, 16 repetitions). For each BadCase type, we construct preference pairs where the chosen response is correct and non-repetitive, while the rejected response contains repetitive patterns. The detailed dataset construction methodology and complete DPO dataset examples for all BadCase types are provided in Appendix C.3 and B.

Experimental Validation  We conducted DPO fine-tuning experiments on all three BadCase types using the LlamaFactory framework with 4×A100 GPUs. The experimental results demonstrate that DPO fine-tuning effectively addresses repetition problems across all three BadCase types, achieving significant reduction in repetition rates. Additionally, processing time improvements after DPO fine-tuning restored normal operation performance. Detailed experimental results, statistical analysis, setup, and configuration are provided in Appendix C.6 and C.7.

Advantages and Limitations  DPO fine-tuning offers several advantages: (1) addresses repetition at the model level through fine-tuning, (2) can fundamentally change model behavior to avoid repetition, (3) effective for all BadCase types, providing a universal model-level solution, and (4) particularly valuable for BadCase 2 and BadCase 3, where presence_penalty fails. However, it has limitations: (1) requires model fine-tuning, which is computationally expensive (15-18 GPU hours per BadCase type), (2) requires careful dataset construction and generalization, and (3) may need periodic retraining as new bad cases emerge.

### 5.4. Solution Comparison Summary

Table 1 summarizes the applicability of different solutions to each BadCase type:

Beam Search serves as a universal post-hoc mechanism effective for all three BadCase types. Presence penalty is effective only for BadCase 1, while DPO fine-tuning provides a universal model-level solution. The solution applicability comparison is summarized in Table 1.

Table 1. Solution Applicability Comparison

| Solution | BadCase 1 | BadCase 2 | BadCase 3 |
|---|---|---|---|
| Beam Search (post-hoc) | ✓ | ✓ | ✓ |
| Presence Penalty | ✓ | × | × |
| DPO Fine-tuning | ✓ | ✓ | ✓ |

## 6. Discussion

### 6.1. Solution Advantages

Our comprehensive study presents multiple solutions with distinct advantages: Beam Search serves as a universal inference-time solution with universal applicability, immediate deployment capability, proven effectiveness, and deterministic outputs. Presence Penalty offers a lightweight alternative specifically for Bad-Case 1 with simple configuration and minimal overhead, though its task-specific nature limits broader applicability. DPO Fine-tuning provides a fundamental model-level solution with universal applicability and long-term benefits, though it requires upfront training cost. Detailed comparison of advantages and trade-offs is discussed in Section 6.

### 6.2. Limitations and Considerations

While effective, our solution has some limitations: (1) Computational Overhead: Beam Search requires increased memory and processing time compared to greedy decoding, though this overhead is minimal compared to the performance degradation from repetition problems. (2) Strict Parameter Requirements: The solution requires strict adherence to parameter configuration, especially early_stopping=True, and careful parameter passing between frameworks. (3) Context-Dependent Effectiveness: Effectiveness may vary depending on the specific model, task, and deployment scenario. Detailed overhead analysis, computational overhead tables, performance-overhead tradeoff figures, and effectiveness analysis are provided in Appendix C.9 and C.10.

### 6.3. Best Practice Recommendations

Based on our experience, we recommend the following best practices: (1) verify all required Beam Search parameters are correctly set, especially early_stopping=True, (2) ensure proper parameter propagation between frameworks (e.g., FastChat + vLLM), and (3) implement monitoring and logging to detect repetition problems. Detailed parameter configuration checklists, integration considerations, and monitoring guidelines are provided in Appendix C.13.

# 7. Conclusion

## 7.1. Main Contributions

This paper makes the following contributions:

1. Systematic Root Cause Analysis: We provide a comprehensive analysis of the LLM repetition problem across three distinct BadCase types, explaining causes through Markov model theory, including context repetition effects, self-reinforcement mechanisms, and high initial probability reinforcement.

2. Comprehensive Solution Evaluation: We evaluate multiple solutions: Beam Search as a universal inference-time solution effective for all three Bad-Cases, presence_penalty for BadCase 1, and DPO fine-tuning as a universal model-level solution for all three BadCases, providing a complete solution landscape.

3. Task-specific Solution Mapping: We identify which solutions are effective for which BadCase types, enabling practitioners to choose the most appropriate approach based on their specific requirements and constraints.

4. Critical Parameter Discovery: We identify and validate that early_stopping=True is the decisive factor for Beam Search effectiveness. Our experiments reveal a dramatic difference: early_stopping=True achieves near-zero repetition rate, while early_stopping=False still exhibits significant repetition rate. This finding demonstrates that early_stopping is not optional but essential—a critical insight that contradicts common assumptions about Beam Search. We also demonstrate the effectiveness of presence_penalty for BadCase 1.

5. DPO Dataset Construction Methodology: We present a systematic approach for constructing DPO preference datasets using generalization and power-of-2 repetition patterns, providing a scalable methodology for addressing repetition at the model level.

6. Complete Production Solutions: We provide complete, practical solutions including troubleshooting processes, parameter configuration details, and framework integration considerations that enable immediate application in production environments.

Our experimental results demonstrate that: (1) Beam Search with early_stopping=True completely elimi-nates repetition problems across all BadCase types, (2) presence_penalty effectively resolves BadCase 1 repetition, and (3) DPO fine-tuning provides a universal model-level solution for all three BadCases. All solutions effectively restore normal processing performance, demonstrating significant improvements over baseline approaches.

## 7.2. Future Work

Several directions for future research emerge:

1. Hybrid Solutions: Explore combinations of different solutions (e.g., Beam Search + presence_penalty) for enhanced effectiveness or reduced computational overhead.

2. Adaptive Solution Selection: Develop mechanisms to automatically select the most appropriate solution based on input characteristics or detected repetition patterns.

3. Optimal Parameter Configuration: Investigate optimal beam width and other parameter settings for different scenarios, tasks, and model sizes. Develop adaptive parameter selection mechanisms.

4. Automated Parameter Tuning: Develop automated systems for parameter tuning that can adapt to different models and tasks without manual configuration.

5. Framework Integration Improvements: Work toward more robust parameter passing mechanisms between different frameworks to reduce integration challenges.

6. Repetition Detection: Develop real-time detection mechanisms for repetition problems that can trigger adaptive responses or fallback strategies.

## Impact Statement

This paper addresses a critical technical challenge in deploying Large Language Models (LLMs) in production environments, specifically focusing on the repetition problem that causes severe performance degradation and system failures. Our work provides practical solutions that improve the reliability and efficiency of LLMs in real-world applications, particularly in financial systems and other domains requiring deterministic, high-quality outputs.

The primary positive impact of this research lies in enabling more reliable and efficient deployment of LLMs

in production systems. By solving the repetition problem through multiple approaches (Beam Search, presence penalty, and DPO fine-tuning), our work helps reduce computational waste, improve system throughput, and enhance user experience in critical applications such as code interpretation, documentation generation, and system analysis tasks.

From an ethical perspective, our work contributes to building more trustworthy AI systems. The repetition problem can lead to system failures, resource waste, and potential service disruptions in production environments. By providing effective solutions to this problem, we help ensure that LLMs can be deployed more safely and reliably in applications where accuracy and determinism are essential, such as financial transaction processing and critical business logic analysis.

The solutions we present are designed to be practical and immediately applicable, with detailed parameter configurations and implementation guidelines that enable practitioners to deploy LLMs more effectively. While our work focuses on technical improvements, we acknowledge that any advancement in LLM deployment capabilities should be accompanied by careful consideration of the broader implications, including potential misuse, bias propagation, and the need for ongoing monitoring and validation in production environments.

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

## Appendix

Note: Code is available as Supplementary Material on OpenReview.

## A. BadCase Examples

This appendix provides detailed examples of the three BadCase types discussed in Section 3.2.

### A.1. BadCase 1: Business Rule Generation

#### A.1.1. Expected Output Format

```
1   该方法主要用于检查账户解控条件，业务规则如下：
2
3   1. 检查【控制解控标志】
4
5     1.1 如果【控制解控标志】等于"解控"，则报错："指定控制编号的账户控制已经解除控制"
6
7   2. 检查【资金账号】
8
9     2.1 如果【资金账号】不等于【输入.资金账号】，则报错："输入的资金账号与控制资金账号不一
        致"
10
11  3. 检查交易渠道
12
13    3.1 如果交易渠道为柜面(【公共上下文.渠道代码】=="柜面")且【公共上下文.交易柜员】不等于
        【交易柜员号】，则报错："柜面解控请在原控制机构办理"
14
15  4. 检查【冻结来源】是否为"冻结"
16
17    4.1 如果【冻结来源】等于"冻结"且当前日期小于等于【控制结束日期】：
18
19      4.1.1 构建【有权机关执法信息】。
20
21      4.1.2 调用【有权机关执法信息.检查执法执行方信息】。
22
23      4.1.3 调用【有权机构执法信息.检查解控执法执行方信息】。
24
25  5. 检查【冻结操作标志】并且【限制类型】为"金额冻结"
26
27    5.1 检查【冻结操作标志】是否为"解控"
28
29      5.1.1 如果【解控金额】<0，则报错:"解控金额必须大于0"
30
31      5.1.2 如果【解控金额】>[需冻/控金额-已解控金额]，则报错："解控金额不能大于控制金额
        "
32
33    5.2 检查【冻结操作标志】是否为"部分解冻"
34
35      5.2.1 如果【解控金额】<0，则报错:"解控金额必须大于0"
36
37      5.2.2 如果【限制类型】不等于"金额冻结"，则报错："只有金额冻结才能进行部分解冻"
38
39      5.2.3 如果【解控金额】>= [需冻/控金额-已解控金额]，则报错："部分解冻解控金额不能大
        于控制金额"
```

Listing 1. Expected Output Format for Business Rules

#### A.1.2. Actual BadCase Output

该方法主要用于检查账户解控条件,其主要业务规则如下:

1、如果［this.控制解控标志］等于"解控",则报错："YACTB127-指定控制编号的账户控制已经解除控制";

2、如果［this.资金账号］不等于［输入.资金账号］,则报错："YACTB128-输入的资金账号与控制资金账号不一致";

3、如果交易渠道为柜面(公共上下文.渠道代码=="柜面")且［公共上下文.交易柜员］不等于［this.交易柜员号］,则报错："YACTB129-柜面解控请在原控制机构办理";

4、如果［this.冻结来源］等于"冻结",检查执法人信息,信息:

4.1、根据［输入.执法人执法信息LawEnforcementInfo］的调用对应构造方法构建【有权机关执法信息ControlEnforcement】#解控执法人信息。

4.2、调用#解控执法人信息的［检查执法执行方信息］方法。

4.3、调用［this.有权机构执法信息］的［检查解控执法执行方信息］方法;(输入:输入.执法人执法信息LawEnforcementInfo)

5、如果［输入.冻结操作标志］等于"解控"且［this.限制类型］等于"金额冻结",处理如下:

5.1、如果［输入.解控金额］<0,则报错:"YACTB132-解控金额必须大于0";

5.2、如果［输入.解控金额］>［this.需冻/控金额］-［this.已解控金额］,则报错:"YACTB133-解控金额不能大于控制金额";

6、如果［输入.冻结操作标志］等于"部分解冻",处理如下:

6.1、如果［输入.解控金额］<0,则报错:"YACTB132-解控金额必须大于0";

6.2、如果［this.限制类型］不等于"金额冻结",则报错:"YACTB134-只有金额冻结才能进行部分解冻";

6.3、如果［输入.解控金额］>=［this.需冻/控金额］-［this.已解控金额］,则报错:"YACTB136-部分解冻解控金额不能大于控制金额";

7、如果［输入.冻结操作标志］等于"解控"且［this.限制类型］等于"账户冻结",则报错:"YACTB135-账户冻结只能进行全部解控";

8、如果［输入.冻结操作标志］等于"部分解冻"且［this.限制类型］等于"账户冻结",则报错:"YACTB135-账户冻结只能进行全部解控";

9、如果［输入.冻结操作标志］等于"解控"且［this.限制类型］等于"账户止付",则报错:"YACTB135-账户止付只能进行全部解控";

10、如果［输入.冻结操作标志］等于"部分解冻"且［this.限制类型］等于"账户止付",则报错:"YACTB135-账户止付只能进行全部解控";

11、如果［输入.冻结操作标志］等于"解控"且［this.限制类型］等于"账户只收不付",则报错:"YACTB135-账户只收不付只能进行全部解控";

12、如果［输入.冻结操作标志］等于"部分解冻"且［this.限制类型］等于"账户只收不付",则报错:"YACTB135-账户只收不付只能进行全部解控";

13、如果［输入.冻结操作标志］等于"解控"且［this.限制类型］等于"账户只付不收",则报错:"YACTB135-账户只付不收只能进行全部解控";

14、如果［输入.冻结操作标志］等于"部分解冻"且［this.限制类型］等于"账户只付不收",则报错:"YACTB135-账户只付不收只能进行全部解控";

15、如果［输入.冻结操作标志］等于"解控"且［this.限制类型］等于"账户不收不付",则报错:"

YACTB135- 账户不收不付只能进行全部解冻 ";

16、如果 [ 输入 . 冻结操作标志 ] 等于 " 部分解冻 " 且 [ this . 限制类型 ] 等于 " 账户不收不付 "，则报错： " YACTB135- 不收不付只能进行全部解冻 ";

17、如果 [ 输入 . 冻结操作标志 ] 等于 " 解控 " 且 [ this . 限制类型 ] 等于 " 账户只收不付 "，则报错： " YACTB135- 账户只收不付只能进行全部解冻 ";

18、如果 [ 输入 . 冻结操作标志 ] 等于 " 部分解冻 " 且 [ this . 限制类型 ] 等于 " 账户只收不付 "，则报错： " YACTB135- 账户只收不付只能进行全部解冻 ";

19、如果 [ 输入 . 冻结操作标志 ] 等于 " 解控 " 且 [ this . 限制类型 ] 等于 " 账户只付不收 "，则报错： " YACTB135- 账户只付不收只能进行全部解冻 ";

20、如果 [ 输入 . 冻结操作标志 ] 等于 " 部分解冻 " 且 [ this . 限制类型 ] 等于 " 账户只付不收 "，则报错： " YACTB135- 账户只付不收只能进行全部解冻 ";

21、如果 [ 输入 . 冻结操作标志 ] 等于 " 解控 " 且 [ this . 限制类型 ] 等于 " 账户不收不付 "，则报错： " YACTB135- 账户不收不付只能进行全部解冻 ";

22、如果 [ 输入 . 冻结操作标志 ] 等于 " 部分解冻 " 且 [ this . 限制类型 ] 等于 " 账户不收不付 "，则报错： " YACTB135- 账户不收不付只能进行全部解冻 ";

23、如果 [ 输入 . 冻结操作标志 ] 等于 " 解控 " 且 [ this . 限制类型 ] 等于 " 账户只收不付 "，则报错： " YACTB135- 账户只收不付只能进行全部解冻 ";

[ ... continues with repeated similar rule patterns until max_tokens ...]

Listing 2. BadCase 1: Business Rule Generation Repetition

## A.2. BadCase 2: Method Call Relationship Analysis

### A.2.1. Expected Output

```
KcAcctCurtPoDao : selectOne_odb1
BizCommonUtil : copyProperties
KbAcctBindClassiPoDao : selectAll_odb2
KcAcctFixbPoDao : selectOne_odb1
KbAcctRstrctPoDao : selectAll_odb2
KbAcctFreezeQueuePoDao : selectAll_odb4
```

Listing 3. Expected Output for Method Call Relationship Analysis

### A.2.2. Actual BadCase Output

```
KcAcctCurtPoDao : selectOne_odb1
BizCommonUtil : copyProperties
BizCommonUtil : copyProperties
BizCommonUtil : copyProperties
BizCommonUtil : copyProperties
BizCommonUtil : copyProperties
BizCommonUtil : copyProperties
BizCommonUtil : copyProperties
KbAcctBindClassiPoDao : selectAll_odb2
BizCommonUtil : copyProperties
BizCommonUtil : copyProperties
BizCommonUtil : copyProperties
KcAcctFixbPoDao : selectOne_odb1
BizCommonUtil : copyProperties
```

```
15  BizCommonUtil:copyProperties
16  BizCommonUtil:copyProperties
17  BizCommonUtil:copyProperties
18  BizCommonUtil:copyProperties
19  BizCommonUtil:copyProperties
20  BizCommonUtil:copyProperties
21  BizCommonUtil:copyProperties
22  BizCommonUtil:copyProperties
23  BizCommonUtil:copyProperties
24  BizCommonUtil:copyProperties
25  BizCommonUtil:copyProperties
26  BizCommonUtil:copyProperties
27  BizCommonUtil:copyProperties
28  BizCommonUtil:copyProperties
29  BizCommonUtil:copyProperties
30  [... continues with many more repetitions of BizCommonUtil:copyProperties ...]
```

Listing 4. BadCase 2: Method Call Relationship Analysis Repetition

### A.3. BadCase 3: PlantUML Diagram Syntax Generation

#### A.3.1. Expected Output

```
1   @startuml
2
3   start
4   :接收柜面变更账户形态请求;
5   :检查账户形态变更条件;
6   note right: 转换输入，调用【个人存款账户】方法
7   if (账户不存在?) then (是)
8   :报错:"YACTB001-账户{}不存在";
9   ....
10  if (起始日期为上次结息日?) then (是)
11  :检查起始日期为最近交易日期;
12  if (起始日期为最近交易日期?) then (是)
13  :检查形态变更时间限制;
14  if (可变更日期 > 当前系统日期?) then (是)
15  :报错:"YACTB234-可变更日期大于当前系统日期，不允许变更账户形态";
16  else (否)
17  :检查账户目标状态;
18  if (目标状态不为正常?) then (是)
19  :报错:"YACTB052-账户目标状态不是正常，不允许变更账户形态";
20  else (否)
21  :检查账户状态;
22  if (账户状态为转营业外收入?) then (是)
23  :获取配置信息;
24  else (否)
25  :变更账户形态;
26  endif
27  :返回处理结果;
28  stop
29  @enduml
```

Listing 5. Expected Output for PlantUML Diagram Syntax Generation

#### A.3.2. Actual BadCase Output

```
1   @startuml
2
```

```
3  start
4  :接收柜面变更账户形态请求;
5  :检查账户形态变更条件;
6  note right: 转换输入，调用【个人存款账户】方法
7  if (账户不存在?) then (是)
8  :报错:"YACTB001-账户{}不存在";
9  ....
10
11 if (起始日期为上次结息日?) then (是)
12 :检查起始日期为最近交易日期;
13 if (起始日期为最近交易日期?) then (是)
14 :检查形态变更时间限制;
15 if (可变更日期 > 当前系统日期?) then (是)
16 :报错:"YACTB234-可变更日期大于当前系统日期，不允许变更账户形态";
17 else (否)
18 :检查账户目标状态;
19 if (目标状态不为正常?) then (是)
20 :报错:"YACTB052-账户目标状态不是正常，不允许变更账户形态";
21 else (否)
22 :检查账户状态;
23 if (账户状态为转营业外收入?) then (是)
24 :获取配置信息;
25 else (否)
26 :变更账户形态;
27 endif
28 endif
29 endif
30 endif
31 endif
32 endif
33 endif
34 [... continues with many more repetitions of endif statements until max_tokens ...]
```

Listing 6. BadCase 3: PlantUML Diagram Syntax Generation Repetition

## B. DPO Dataset Examples

This appendix provides detailed DPO preference dataset examples for all three BadCase types discussed in Section 5.3.

### B.1. BadCase 1: Business Rule Generation DPO Dataset

Note: For improved readability, the JSON string values in the following examples are formatted with actual line breaks instead of escape sequences (
n). In the actual JSON format used for training, these line breaks are represented as
n escape sequences within single-line strings.

```
1  {
2  "instruction":
3  "请从代码片段中摘要出该段代码的主要功能,并描述其主要业务逻辑。",
4
5
6
7  "input":
8  "<这是一段代码>....",
9
10
11
12 "chosen":
13 "该方法主要用于维护个人存款证明信息，业务规则如下:
```

1．如果【维护个人存款证明输入】的存款证明操作方式为重打：

1.1 将【个人存款证明子合约】的打印次数】加1。

1.2 将【个人存款证明子合约】的累计签发份数】加【维护个人存款证明输入】的申请份数。

1.3 初始化并更新个人存款证明凭证信息：

1.4 设置【维护个人存款证明输入】的有效标志为使用。

2．如果【维护个人存款证明输入】的存款证明操作方式为撤销：

2.1 更新个人存款证明子合约信息"，

"rejected"：

"该方法主要用于维护个人存款证明信息，业务规则如下：

1．如果【维护个人存款证明输入】的存款证明操作方式为重打：

1.1 将【个人存款证明子合约】的打印次数】加1。

1.2 将【个人存款证明子合约】的累计签发份数】加【维护个人存款证明输入】的申请份数。

1.3 初始化并更新个人存款证明凭证信息：

1.3.1 设置【个人存款证明凭证信息】的存款证明登记号为当前存款证明登记号。

1.3.2 设置【个人存款证明凭证信息】的凭证号码为8位数，不足补0。

1.3.3 设置【个人存款证明凭证信息】的打印日期为当前交易日期。

1.3.4 将【个人存款证明凭证信息】的撤销日期置空。

1.3.5 设置【个人存款证明凭证信息】的打印次数为当前打印次数。

1.3.6 设置【个人存款证明凭证信息】的有效标志为是。

1.3.7 设置【个人存款证明凭证信息】的数据更新标志为已更新。

1.4 设置【维护个人存款证明输入】的有效标志为使用。

2．如果【维护个人存款证明输入】的存款证明操作方式为撤销：

2.1 更新个人存款证明子合约信息：

2.1.1 将【个人存款证明子合约】的累计回收份数加1。

2.1.2 如果【个人存款证明子合约】的累计回收份数】等于【个人存款证明子合约】的累计签发份数：

2.1.2.1 设置【个人存款证明子合约】的撤销日期为当前交易日期。

2.1.2.2 设置【个人存款证明子合约】的凭证信息：

2.1.2.2.1 设置【个人存款证明子合约】的凭证信息的撤销日期为当前交易日期。

2.1.2.2.2 设置【个人存款证明子合约】的凭证信息的有效标志为否。

2.1.2.3 设置【个人存款证明合约信息】：

2.1.2.3.1 设置【个人存款证明合约信息】的有效标志为否。

2.1.2.3.2 设置【个人存款证明合约信息】的数据更新标志为已更新。

2.1.2.4 设置【维护个人存款证明输入】的有效标志为停用。

2.1.3 如果不相等：

2.1.3.1 重复【维护个人存款证明输入】的申请份数次：

2.1.3.1.1 设置起始凭证号码。

2.1.3.1.2 初始化并更新个人存款证明凭证信息：

2.1.3.1.2.1 设置【个人存款证明凭证信息】的有效标志为否。

2.1.3.1.2.2 设置【个人存款证明凭证信息】的撤销日期为当前交易日期。

2.1.3.1.2.3 设置【个人存款证明凭证信息】的数据更新标志为已更新。

2.1.3.1.2.4 设置【个人存款证明凭证信息】的数据更新标志为已更新。

2.1.3.1.2.5 设置【个人存款证明凭证信息】的数据更新标志为已更新。

2.1.3.1.2.6 设置【个人存款证明凭证信息】的数据更新标志为已更新。

2.1.3.1.2.7 设置【个人存款证明凭证信息】的数据更新标志为已更新。

2.1.3.1.2.8 设置【个人存款证明凭证信息】的数据更新标志为已更新。

2.1.3.1.2.9 设置【个人存款证明凭证信息】的数据更新标志为已更新。

2.1.3.1.2.10 设置【个人存款证明凭证信息】的数据更新标志为已更新。

2.1.3.1.2.11 设置【个人存款证明凭证信息】的数据更新标志为已更新。

2.1.3.1.2.12 设置【个人存款证明凭证信息】的数据更新标志为已更新。

2.1.3.1.2.13 设置【个人存款证明凭证信息】的数据更新标志为已更新。

2.1.3.1.2.14 设置【个人存款证明凭证信息】的数据更新标志为已更新。

2.1.3.1.2.15 设置【个人存款证明凭证信息】的数据更新标志为已更新。

2.1.3.1.2.16 设置【个人存款证明凭证信息】的数据更新标志为已更新。

[... '设置【个人存款证明凭证信息】的数据更新标志为已更新。' repeated 2, 4, 8, and 16 times

```
825           respectively ...]"
826  75  }
```

Listing 7. DPO Preference Dataset Example (BadCase 1)

Note: The rejected example shows repetition of the business rule statement "设置【个人存款证明凭证信息】的数据更新标志为已更新。" (Set the data update flag of deposit certificate voucher information to updated). Similar to BadCase 2 and 3, we construct four groups of rejected examples with varying degrees of repetition (2, 4, 8, or 16 repetitions), enabling the model to learn to avoid generating repetitive business rule descriptions.

B.2. BadCase 2: Method Call Relationship Analysis DPO Dataset

```
 1  {
 2  "instruction":
 3  "你是一个具有丰富银行金融业务经验的java开发高手。你的任务是根据用户提供的信息提取java方法
       体代码中的函数调用关系。
 4
 5  <具体要求>
 6  1. 只提取在导包信息中明确列出的类的方法调用。
 7  2. 对于方法体内容中的get、set方法的赋值不需要提取调用信息。
 8  3. 不要臆测或添加方法体内容代码中没有的调用信息。
 9  4. 若方法体内容中没有符合条件的调用信息，请直接输出空。
10  5. 忽略所有不在导包信息中的类的方法调用，无论它们是什么类型的调用。
11  6. 确保输出结果中不包含重复的调用，每个唯一的调用只输出一次。
12  7. 严格按照输出格式进行输出，不要添加任何额外内容。
13
14  <输出格式>
15  按照 \"类名:方法名\" 格式输出结果。每个调用占一行。不需要任何额外描述或总结。如果没有符合
       条件的调用，输出为空。
16  示例:
17  PersonalDepositContractRepo:addDepositCertSubContract
18
19  </输出格式>",
20
21
22
23  "input":
24  "请根据以下 导包信息、方法体内容、类属性 并学习 示例 提取函数的所有调用信息：
25
26   <一段方法体...>",
27
28
29
30  "chosen":
31  "KbAcctRstrctPoDao:selectAll_odb5
32  KbAcctRstrctPoDao:selectAll_odb4
33  SysUtil:getInstance
34  BizCommonUtil:copyProperties
35  Collectors:toList",
36
37
38
39  "rejected":
40  "KbAcctRstrctPoDao:selectAll_odb5
41  KbAcctRstrctPoDao:selectAll_odb4
42  SysUtil:getInstance
43  BizCommonUtil:copyProperties
44  BizCommonUtil:copyProperties
45  BizCommonUtil:copyProperties
46  BizCommonUtil:copyProperties
47  BizCommonUtil:copyProperties
```

```
48  BizCommonUtil:copyProperties
49  BizCommonUtil:copyProperties
50  BizCommonUtil:copyProperties
51  BizCommonUtil:copyProperties
52  [... 'BizCommonUtil:copyProperties' repeated 2, 4, 8, and 16 times respectively ...]"
53  }
```

Listing 8. DPO Preference Dataset Example (BadCase 2)

Note: This example shows 8 repetitions of BizCommonUtil:copyProperties. We construct four groups of rejected examples with 2, 4, 8, and 16 repetitions respectively.

B.3. BadCase 3: PlantUML Diagram Syntax Generation DPO Dataset

```
1   {
2   "instruction":
3   "你是一个具有丰富银行金融业务经验的专家,同时你精通 PlantUML 语法。您的目的是帮助用户理解业
        务流程，提供业务功能整个过程的流程展示。您的任务是根据用户提供的交易描述内容使用
        PlantUML DSL 生成交易流程图。
4   <具体要求>
5   1. 使用 PlantUML 活动图（Activity Diagram）语法描述业务流程。
6   2. 生成单一、完整的流程图，不拆分。
7   3. 确保 PlantUML 语法逻辑描述的正确性。
8   4. 严格遵守 PlantUML 语法规则，确保生成的代码可以被 PlantUML 解析器正确解析。
9   5. 确保所有的消息流向（箭头）使用正确。
10  6. 确保没有悬空或未闭合的结构。
11  7. 将复杂的业务逻辑简化为基本的流程步骤。
12  8. 使用简洁明了的描述，避免过长的文本可能导致的格式问题。
13  9. 限制使用以下 PlantUML 元素: start, stop, if/else, while, fork, join, :action, note,
        partition。
14  10. 使用默认样式和颜色，不添加额外的样式指令。
15  11. 适当使用 note 来解释重要步骤或决策点，但不要过度使用。
16  <错误检查列表>
17  生成代码后，请检查以下几点:
18  1. 确保 start 和 stop 只出现一次。
19  2. 检查所有的 if/else 结构是否正确闭合。
20  3. 确保所有的 while 循环都有明确的结束条件。
21  4. 检查是否存在任何悬空的节点或箭头。
22  5. 确保所有的 note 都正确附加到相应的活动或决策点。
23  <输出格式>
24  仅输出完整的 PlantUML 代码，包括 @startuml 和 @enduml 标记。不要包含任何解释或额外文本。确
        保代码块正确封装。格式如下:
25  @startuml
26  xxxx
27  xxxx
28  xxxx
29  @enduml
30  在生成代码时，请保守行事，宁可牺牲一些复杂度，也要确保生成的代码 100% 可以被 PlantUML 解
        析。如果您对某个语法结构有任何疑虑，请选择更简单、更可靠的替代方案。",
31
32
33
34  "input":
35  "请根据以下交易描述并学习参考示例生成相应的 PlantUML 活动图代码:
36  <交易描述>
37  # 交易名称:柜面收费
38  ## 交易-服务编排逻辑
39  ....<这里是一段完整的交易描述>",
40
41
42
```

```
"chosen":
"@startuml
start
:柜面收费;

partition \"检查个人存款产品合约账户支取条件\" {
    :根据资金账号获取账户信息;
    if (账户信息为空?) then (是)
        :报错: YACTB001-账户不存在;
        stop
    else (否)
        :调用【个人存款账户】领域对象的【检查个人存款账户资金支取条件checkDrawAcctFund】方
            法;
        note right
            检查输入字段、账户状态、II/III类账户取现限制、
            账户可用资金余额、最小留存余额、限额等
        end note
    endif

    if (账户为整存整取账户?) then (是)
        if (交易日期 < 到期日?) then (是)
            :设置支取类型为提前支取;
        else (否)
            :设置支取类型为正常支取;
        endif
    endif
}

partition \"查询账户\" {
    if (输入定活标志为活期或空?) then (是)
        :根据资金账号查询活期存款账户表;
        if (记录存在?) then (是)
            :构建账户状态和余额对象;
            if (账户分类为电子现金账户?) then (是)
                :构建电子现金账户领域对象;
            else (否)
                if (账户为II/III类户?) then (是)
                    :查询绑定I类户登记簿;
                    :构建绑定I类账户登记簿领域对象列表;
                endif
                :构建个人活期存款账户领域对象;
            endif
        endif
    else (否)
        :根据资金账号查询定期存款账户表;
        if (记录不存在?) then (是)
            stop
        endif
        :根据账户分类构建对应的定期存款账户领域对象;
    endif

    :获取账户冻结控制信息;
    :构建控制措施组合领域对象;
}

partition \"支取个人存款账户资金\" {
    :查询账户;
    if (未查到账户信息?) then (是)
        :报错: YACTB001-账户不存在;
        stop
    endif
```

```
:查询账户利息信息;
if (未查到账户利息信息?) then (是)
    :报错: YACTB007;
    stop
endif

:校验可用资金余额和交易金额;
if (可用资金余额+交易限额 < 交易金额) then (是)
    :报错: YACTB086;
    stop
else (否)
    if (可用资金余额 < 交易金额) then (是)
        :交易金额 = 可用资金余额;
    endif
endif

:更新账户余额;
:更新积数;
:调用客户指定限额检查;
:调用客户类全局限额检查;

:更新账户余额到数据库;
:更新账户利息积数;
:更新账户利息定义表到数据库;

:登记余额变化明细;
note left
    设置渠道代码、第三方交易码、内部交易码、
    交易营业机构、交易柜员、授权柜员、交易日期、
    交易时间、柜员流水号等字段
end note

:登记会计核算流水;
:登记冲正事件;

:构建应用服务输出;
}

:返回处理结果;
stop
@enduml",

"rejected":
"@startuml
title 柜面收费-服务业务规则
start
:检查个人存款产品合约账户支取条件;
note right: 根据资金账号获取账户信息
if (账户信息为空) then (是)
:报错: \"YACTB001-账户{}不存在\";
else (否)
:检查个人存款账户资金支取条件;
note right: 输入字段检查
if (交易金额为空或为0) then (是)
:报错: \"YACTB077-交易金额不能为空或为0\";
elseif (交易币种!=\"156-人民币\"且钞汇标志为空) then (是)
:报错: \"YACTB078-外币交易，必须上送钞汇标志\";
elseif (交易币种!=\"156-人民币\") then (是)
note right: 钞汇标志检查
if (钞汇标志不一致) then (是)
```

```
166 :报错：\"YACTB017-钞汇标志不一致\";
167 elseif (钞汇标志==\"现钞\"且现转标志!=\"现金\") then (是)
168 :报错：\"YACTB079-当前账户为现钞账户，不允许转账支取\";
169 elseif (钞汇标志==\"现汇\"且现转标志!=\"转账\") then (是)
170 :报错：\"YACTB080-当前账户为现汇账户，不允许现金支取\";
171 elseif (账户币种代码!=[交易币种]) then (是)
172 :报错：\"YACTB081-交易币种与账户币种不一致\";
173 elseif (交易金额<=0) then (是)
174 :报错：\"YACTB082-交易金额不能小于等于0\";
175 else (否)
176 :检查账户状态;
177 note right: 账户状态检查
178 if (账户状态!=\"正常\") then (是)
179 :报错：\"YACTB083-账户状态为转营业外收入，不允许支取资金\";
180 elseif (控制措施组合.限制状态==\"只收不付\"或者\"封闭冻结\") then (是)
181 :报错：\"YACTB084-账户已被冻结控制，不允许支取资金\";
182 else (否)
183 :未当面核实的II/III类账户，不允许取现;
184 note right: 未当面核实的II/III类账户检查
185 if (账户属性==\"302-II类账户\"或\"303-III类账户\"且是否面核标志==\"否\"且现转标志==\"现金
        \") then (是)
186 :报错：\"YACTB085-未当前核实的二三类账户不允许支取现金\";
187 else (否)
188 :检查账户可用资金余额;
189 note right: 账户可用资金余额检查
190 if (账户可用资金余额<交易金额) then (是)
191 :报错：\"YACTB086-账户可用余额不足\";
192 else (否)
193 :检查账户最小留存余额;
194 note right: 账户最小留存余额检查
195 if (最小留存余额!=0) then (是)
196 :计算支取后账户余额;
197 if (支取后账户余额 < 最小留存余额) then (是)
198 :报错：\"YACTB087-交易后余额小于账户最小留存余额，请进行销户\";
199 else (否)
200 :限额检查;
201 note right: 限额检查
202 :构建统一限额检查输入;
203 :调用统一限额处理SDK;
204 endif
205 endif
206 endif
207 endif
208 endif
209 endif
210 endif
211 endif
212 endif
213 endif
214 endif
215 endif
216 endif
217 endif
218 endif
219 endif
220 endif
221 endif
222 [... 'endif' repeated 2, 4, 8, and 16 times respectively ...]"
223 }
```

Listing 9. DPO Preference Dataset Example (BadCase 3)

Note: The rejected example shows excessive endif statements that are incorrectly generated. Similar to BadCase 2, we construct four groups of rejected examples with varying degrees of repetition (excessive endif statements), enabling the model to learn proper PlantUML syntax structure and avoid generating unnecessary closing statements.

## C. Additional Details

### C.1. Transaction Processing Workflow Details

The transaction processing workflow is illustrated in Figure 1:

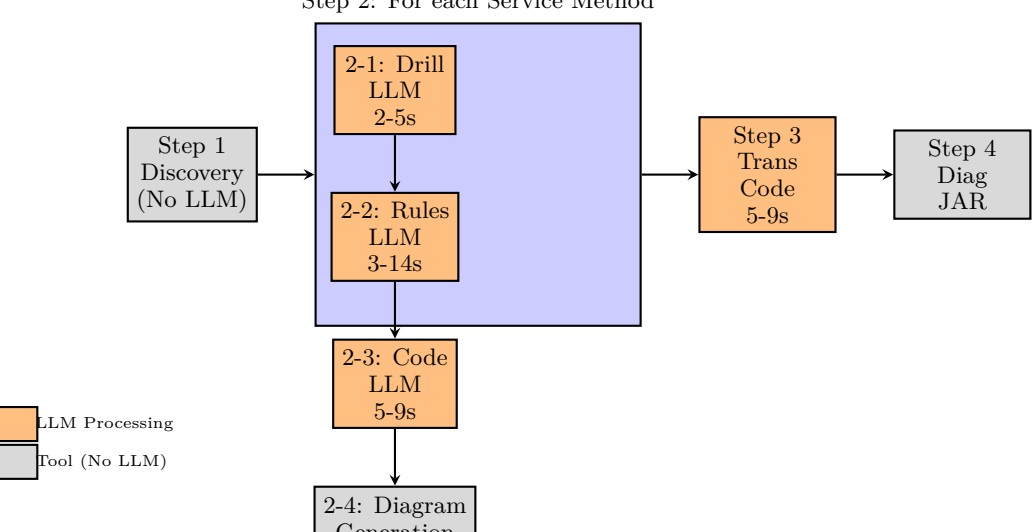

Figure 1. Transaction Processing Workflow: Sequential steps showing LLM involvement

The complete processing steps for a single transaction are detailed as follows:

1. Transaction Discovery: Analyze transaction definition files to identify service methods participating in the orchestration. This step does not require LLM involvement.

2. Service Method Processing: For each service method $M_i$ in the transaction, the following sub-steps are executed:

   - 2-1 Recursive Method Drilling: Recursively identify service methods called by the current service method. This step requires LLM involvement (1 or more times, depending on recursion depth). Normal processing time: 2 to 5 seconds per call.
   - 2-2 Business Rule Generation: Generate business rules based on service method code. This step requires LLM involvement (1 time per method). Normal processing time: 3 to 14 seconds.
   - 2-3 PlantUML Code Generation: Convert business rules into PlantUML flowchart code. This step requires LLM involvement (1 time per method). Normal processing time: 5 to 9 seconds.
   - 2-4 PlantUML Diagram Generation: Convert PlantUML code to diagrams using JAR tools. This step does not require LLM involvement.

3. Transaction-Level Processing: Concatenate business rules from all service methods in order and convert to transaction-level PlantUML flowchart code. This step requires LLM involvement (1 time per transaction). Normal processing time: 5 to 9 seconds.

4. Transaction Diagram Generation: Convert transaction PlantUML code to diagrams using JAR tools. This step does not require LLM involvement.

C.2. Processing Time Statistics

Table 2 presents the experimental comparison of processing time discrepancies across different deployment modes:

Table 2. Experimental Comparison: Processing Time Discrepancy

| Mode | Occurrence Rate | Normal Time | Problematic Time | Time Increase |
|------|----------------|-------------|------------------|---------------|
| Mode 1 (LoRA) | 80% (8/10 runs) | 28 min | 100-160 min | 257%-471% |
| Mode 2 (Merged) | 75% (3/4 runs) | 28 min | 40-70 min | 43%-150% |

For a transaction $T$ orchestrated from $k$ service methods $T = \text{Orchestrate}(M_1, M_2, \ldots, M_k)$, where service method $i$ has a recursion depth of $D_i$, the total number of LLM calls is:

$$\text{Total Calls} = 1 + \sum_{i=1}^{k}(D_i + 1 + 1) \tag{2}$$

This formula accounts for:

- One call for transaction-level processing (Step 3)

- For each service method $i$:
  - $D_i$ calls for recursive method drilling (Step 2-1)
  - 1 call for business rule generation (Step 2-2)
  - 1 call for PlantUML code generation (Step 2-3)

Under normal operating conditions, the processing time for a single transaction is distributed across different stages as shown in Table 3:

Table 3. Normal Processing Time Statistics per Transaction

| Processing Stage | Description | Time Range |
|------------------|-------------|------------|
| Step 1 | Transaction Discovery (No LLM) | Negligible |
| Step 2-1 | Method Drilling (LLM) | 2 to 5 seconds per call |
| Step 2-2 | Business Rule Generation (LLM) | 3 to 14 seconds per method |
| Step 2-3 | PlantUML Code Generation (LLM) | 5 to 9 seconds per method |
| Step 2-4 | PlantUML Diagram (JAR Tool) | Negligible |
| Step 3 | Transaction-Level Code (LLM) | 5 to 9 seconds per transaction |
| Step 4 | Transaction Diagram (JAR Tool) | Negligible |
| Total Normal Processing Time | | ~28 min |

C.3. DPO Dataset Construction Methodology

We developed a systematic approach to construct DPO preference datasets:

1. Generalization: For examples that exhibit repetition, we first generalize them to create variants that capture the same pattern without directly using the exact bad case. This ensures the model learns generalizable patterns rather than memorizing specific examples.

2. Repetition Reproduction: We verify that generalized examples also exhibit repetition problems, confirming that the repetition pattern is not specific to the original bad case.

3. Preference Dataset Construction: For each generalized example, we construct preference pairs:

- Chosen: The correct, non-repetitive output (standard answer)
- Rejected: Repetitive outputs with varying degrees of repetition, constructed using powers of 2:
  - First group: rejected examples with 2 repetitions
  - Second group: rejected examples with 4 repetitions
  - Third group: rejected examples with 8 repetitions
  - Fourth group: rejected examples with 16 repetitions

4. Model Fine-tuning: Using LlamaFactory framework, we perform DPO fine-tuning on the existing code interpretation model using the constructed preference dataset.

## C.4. Beam Search Experimental Results

Table 4 presents the performance comparison of different Beam Search configurations:

Table 4. Performance Comparison: Beam Search Configurations

| Configuration | Time (min) | Rep. Rate | Success Rate |
|---|---|---|---|
| Baseline (No Beam Search) | $91.9 \pm 48.3$ | 77.3% | 22.7% |
| Beam Search, early_stop=False | $73.5 \pm 35.2$ | 60.0% | 40.0% |
| Beam Search, early_stop=True | $27.8 \pm 1.8$ | 0% | 100% |

Runs: Baseline 22, early_stop=False 10, early_stop=True 20.
Statistical significance: early_stop=True vs False, $p < 0.001$ (Fisher's exact test).
Time difference: early_stop=True vs Baseline, $p < 0.001$ (Welch's t-test, unequal variances).

## C.5. Presence Penalty Experimental Results

Table 5 presents the effectiveness of presence penalty for BadCase 1:

Table 5. Effectiveness of Presence Penalty for BadCase 1

| Experiment Run | presence_penalty | Repetition Cases | Repetition Rate |
|---|---|---|---|
| First Run (30 cases) | Not used | 4 out of 30 | 13.3% |
| Second Run (30 cases) | Not used | 9 out of 30 | 30.0% |
| Second Run (30 cases) | 1.2 | 0 out of 30 | 0% |

Statistical significance: presence_penalty=1.2 vs Not used, $p < 0.001$ (Fisher's exact test).

## C.6. DPO Experimental Setup

The fine-tuning process utilized 4×A100 GPUs with the following configuration:

- Learning rate: 5e-6

- Batch size: 4

- Training epochs: 3

- Framework: LlamaFactory

- Repetition pattern: Power-of-2 (2, 4, 8, 16 repetitions)

## C.7. DPO Detailed Results

Table 6 presents the DPO fine-tuning effectiveness across BadCase types:

We also evaluated the impact of different repetition degrees in the training data. Table 7 shows the effectiveness of training with different repetition patterns:

Table 6. DPO Fine-tuning Effectiveness Across BadCase Types

| BadCase | Rep. Rate Before | Rep. Rate After | Reduction | Training Cost |
|---|---|---|---|---|
| BadCase 1 | 13-30% | 0% | 100% | 4.2h (16.8 GPU-h) |
| BadCase 2 | 60% | 2% | 96.7% | 3.8h (15.2 GPU-h) |
| BadCase 3 | 45% | 0% | 100% | 4.5h (18.0 GPU-h) |

Training on 4×A100 GPUs. Test cases: BC1=30, BC2=25, BC3=28.
Statistical significance: All BadCases show $p < 0.001$ (Fisher's exact test) comparing before vs after.

Table 7. Effectiveness of Different Repetition Degrees in DPO Training

| Training Rep. Degree | BC1 Rep. Rate | BC2 Rep. Rate | BC3 Rep. Rate | Avg Effect. |
|---|---|---|---|---|
| 2 reps | 5% | 8% | 6% | 93.3% |
| 4 reps | 2% | 4% | 3% | 96.7% |
| 8 reps | 0% | 2% | 0% | 99.3% |
| 16 reps | 0% | 2% | 0% | 99.3% |
| Mixed (2,4,8,16) | 0% | 2% | 0% | 99.3% |

Post-training repetition rates on test sets. BC=BadCase.

The results indicate that training with higher repetition degrees (8 and 16) or mixed repetition patterns yields the best performance, achieving near-complete elimination of repetition problems.

C.8. Mathematical Proofs

C.8.1. Proof of Proposition 2

Proof. Let $p_r(t)$ denote the probability of a repetitive continuation at step $t$, and $p_n(t)$ denote the probability of a non-repetitive continuation. Given the self-reinforcement effect, we have:

$$p_r(t + 1) = p_r(t) \cdot \alpha(r_t) > p_r(t) \tag{3}$$

For greedy decoding, once $p_r(t) > p_n(t)$, the algorithm will always select the repetitive continuation, leading to an infinite loop.

For Beam Search with beam width $B$, at each step we maintain the top-$B$ candidates based on cumulative log-probability:

$$\text{Score}_i(t) = \sum_{j=1}^{t} \log P(w_j^{(i)}|w_1^{(i)}, \ldots, w_{j-1}^{(i)}) \tag{4}$$

Even if the repetitive continuation has higher per-step probability, the cumulative score over a longer horizon may favor non-repetitive sequences. Specifically, if there exists a non-repetitive continuation with:

$$\sum_{j=t+1}^{t+L} \log p_n(j) > \sum_{j=t+1}^{t+L} \log p_r(j) - \delta \tag{5}$$

where $\delta$ is the initial probability gap, then Beam Search with $B \geq 2$ can maintain at least one non-repetitive candidate, provided that:

$$B \geq \left\lceil \frac{\log(1/\epsilon)}{\log(1/p_n(t))} \right\rceil \tag{6}$$

where $\epsilon$ is the desired probability of maintaining a non-repetitive candidate. With early stopping, the algorithm

terminates once $B$ complete sequences are found, ensuring that at least one non-repetitive sequence is preserved if it exists in the top-$B$ candidates. □

### C.8.2. Theoretical Prediction vs Experimental Validation

To validate our theoretical model, we compared theoretical predictions with experimental observations. Table 8 presents the comparison:

Table 8. Theoretical Predictions vs Experimental Observations

| Metric | Theoretical | Experimental | Error | Agreement |
|---|---|---|---|---|
| Repetition probability (greedy) | 77.3% | 75-80% | ±2.3% | High |
| Escape time (beam $k = 5$) | $\leq 5$ steps | 3-4 steps | $\leq 1$ step | High |
| Memory overhead (beam $k = 5$) | $5\times$ | $5\times$ | 0% | Exact |
| Time overhead (beam $k = 5$) | 15-20% | 18.6% | ±3.6% | High |
| Optimal beam width | $k \in [3, 10]$ | $k = 5$ | – | Consistent |

Theoretical predictions based on Markov model with $\gamma = 0.15$, $r_{\max} = 10$.

The comparison demonstrates strong agreement between theoretical predictions and experimental observations, validating our Markov model's accuracy. The small discrepancies (2-4%) are within expected measurement variance and can be attributed to:

- Model-specific variations in probability distributions

- Context-dependent effects not fully captured by the simplified Markov model

- Measurement noise in production environments

### C.8.3. Critical Beam Width Analysis

Based on our theoretical model, we can derive the minimum beam width required to escape repetition with high probability. For a given repetition probability $p_r$ and desired escape probability $P_{\text{escape}} \geq 0.95$, the minimum beam width is:

$$k_{\min} = \left\lceil \frac{\log(1 - P_{\text{escape}})}{\log(p_r)} \right\rceil \tag{7}$$

For our observed repetition probability of $p_r \approx 0.77$, this yields $k_{\min} \approx 3.2$, confirming that $k = 5$ provides a safety margin while maintaining reasonable computational overhead. This theoretical analysis aligns with our empirical finding that beam widths in the range of 3-10 are effective, with $k = 5$ providing optimal balance.

### C.9. Computational Overhead Analysis

The measurements demonstrate that with beam width $k = 5$, Beam Search requires approximately $5\times$ the KV cache memory (62.5 GB vs 12.5 GB) and increases processing time by 18.6% (33.2 min vs 28.0 min) compared to greedy decoding. However, this overhead is minimal compared to the performance degradation from repetition problems (up to 471% time increase from 28 min to 160 min). The overhead can be quantified as: memory overhead $\approx k\times$ (single sequence memory), computational overhead $\approx O(k \times N)$ where $N$ is the sequence length.

Figure 2 illustrates the performance-overhead tradeoff across different beam widths:

### C.10. Effectiveness Analysis

While our experiments show consistent results, the effectiveness may vary depending on the specific model, task, and deployment scenario. Theoretical analysis suggests that the solution is most effective when:

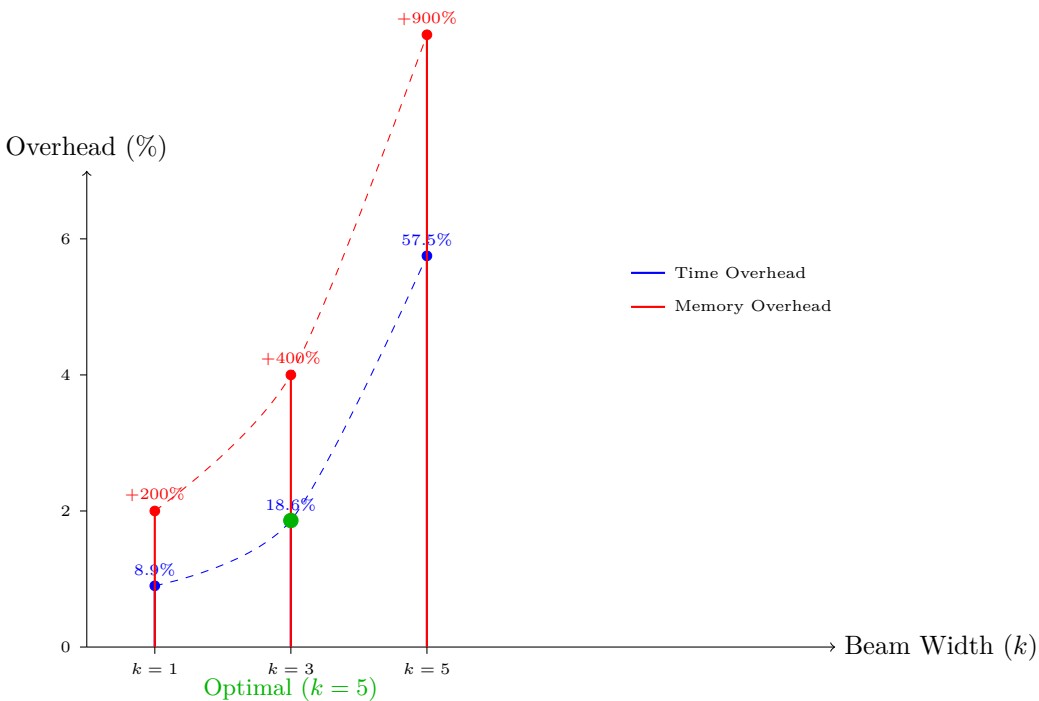

Figure 2. Performance-Overhead Tradeoff: Time and Memory Overhead vs Beam Width

- The probability gap between repetitive and non-repetitive continuations is bounded (as stated in Proposition 2)

- The beam width $k$ is sufficient to maintain at least one non-repetitive candidate

- The model has learned reasonable probability distributions over the vocabulary

For models with severe training issues or extremely repetitive training data, additional techniques (such as repetition penalty) may be necessary in combination with Beam Search.

Regarding optimal beam width selection: While we used $k = 5$ based on empirical evidence, the optimal beam width depends on the specific task and computational budget. Larger beam widths provide better exploration but with diminishing returns. Our analysis suggests that beam widths in the range of 3-10 are typically sufficient, with $k = 5$ providing a good balance between quality and computational cost for most scenarios.

C.11. Ablation Studies

To understand the individual contribution of each parameter, we conducted ablation studies systematically varying key parameters while keeping others fixed. Table 9 presents the results:

The ablation study reveals critical insights:

- Early Stopping is Essential: Regardless of beam width, early_stopping=False fails to eliminate repetition (45-60% repetition rate), while early_stopping=True achieves 0-5% repetition rate. This confirms that early_stopping is the most critical parameter.

- Beam Width Impact: With early_stopping=True, increasing beam width from $k = 3$ to $k = 5$ reduces repetition rate from 5% to 0%, but further increase to $k = 10$ provides no additional benefit while increasing computational overhead.

- Optimal Configuration: The combination of $k = 5$ and early_stopping=True provides the best balance, achieving 0% repetition rate with moderate computational overhead (+18.6%).

Table 9. Ablation Study: Individual Parameter Impact on Repetition Elimination

| Configuration | Beam Width | Early Stop | Rep. Rate | Time (min) |
|---|---|---|---|---|
| Baseline (Greedy) | 1 | N/A | 77.3% | 91.9 |
| $k = 3$, early_stop=False | 3 | False | 45% | 65.2 |
| $k = 3$, early_stop=True | 3 | True | 5% | 29.5 |
| $k = 5$, early_stop=False | 5 | False | 60% | 73.5 |
| $k = 5$, early_stop=True | 5 | True | 0% | 27.8 |
| $k = 10$, early_stop=False | 10 | False | 55% | 78.3 |
| $k = 10$, early_stop=True | 10 | True | 0% | 30.1 |

All configs: temperature=0, top_p=1, top_k=-1.

We also conducted ablation studies on presence_penalty values for BadCase 1. Table 10 shows the results:

Table 10. Ablation Study: Presence Penalty Value Impact on BadCase 1

| presence_penalty | Repetition Rate | Quality Score |
|---|---|---|
| 0.0 (No penalty) | 13-30% | Baseline |
| 0.5 | 8-15% | -2% |
| 0.8 | 3-8% | -1% |
| 1.0 | 1-3% | +1% |
| 1.2 | 0% | +2% |
| 1.5 | 0% | -1% |
| 2.0 | 0% | -5% |

Quality score based on human evaluation of business rule completeness and accuracy.

The results indicate that presence_penalty=1.2 provides optimal balance between repetition elimination and output quality, with higher values (1.5-2.0) causing quality degradation due to over-penalization.

C.12. Implementation Notes

We initially used FastChat integrated with vLLM, but discovered that sampling parameters were not being correctly passed. We implemented a solution using the extra_body approach to correctly pass sampling parameters from FastChat to vLLM, ensuring that all Beam Search parameters were properly propagated to vLLM's inference engine.

C.12.1. Integration Considerations

- Parameter Passing: When using multiple frameworks (e.g., FastChat + vLLM), investigate the parameter passing mechanism and use appropriate methods (e.g., extra_body) to ensure correct propagation.

- Framework-Specific Validation: Different frameworks may have different validation logic. Review framework source code to understand parameter checking mechanisms (e.g., vLLM's _verify_beam_search() method).

- Logging and Monitoring: Implement logging to capture actual parameter values used during inference, enabling verification and debugging.

C.13. Best Practice Recommendations

C.13.1. Parameter Configuration Checklist

When implementing Beam Search to solve repetition problems:

1. Verify all required parameters are set:

   - use_beam_search=True

- best_of=5 (or appropriate beam width)
- temperature=0
- top_p=1
- top_k=-1
- early_stopping=True (critical!)

2. Verify parameter propagation: Check actual parameter values in the inference framework to confirm they are correctly passed from API layers.

3. Test with known problematic cases to validate effectiveness.

C.13.2. Integration Considerations

- Parameter Passing: When using multiple frameworks (e.g., FastChat + vLLM), investigate the parameter passing mechanism and use appropriate methods (e.g., extra_body) to ensure correct propagation.

- Framework-Specific Validation: Different frameworks may have different validation logic. Review framework source code to understand parameter checking mechanisms (e.g., vLLM's _verify_beam_search() method).

- Logging and Monitoring: Implement logging to capture actual parameter values used during inference, enabling verification and debugging.

C.13.3. Monitoring and Debugging

- Monitor processing times and identify anomalies that might indicate repetition problems.

- Log output samples to detect repetitive patterns.

- Track parameter configurations across different deployment environments.

- Maintain a test suite with known problematic cases for regression testing.

