# OpenReview forum: "Solving LLM Repetition Problem in Production:A Comprehensive Study of Multiple Solutions"
_ICML.cc/2026/Conference — Submitted to ICML 2026_

### Official Review · Reviewer_WZw6 · 2026-03-10

**Soundness:** 2
**Presentation:** 3
**Significance:** 3
**Originality:** 3
**Overall Recommendation:** 3
**Confidence:** 4

**Summary:**

The paper investigates a persistent Large Language Model repetition problem observed in a real financial‑transaction code‑analysis system, where the model would enter deterministic loops and repeatedly generate the same content until hitting its maximum token limit.The authors investigate this behavior in the context of a financial transaction code‑interpretation pipeline, where LLMs sequentially analyze service methods, generate business rules, extract method relationships, and produce PlantUML diagrams.These tasks consistently triggered repetitive outputs such as redundant conditional blocks, repeatedly emitted method names, or runaway endif statements.

To address this, the authors combine theoretical and empirical analysis to evaluate three classes of solutions. They show that Beam Search can eliminate repetition only when correctly parameterized particularly when early_stopping=True, which they identify as the decisive factor enabling the search to escape repetitive trajectories. They also test presence‑penalty tuning, which reduces repetition in business‑rule generation but does not generalize to the other use cases. Finally, they apply Direct Preference Optimization (DPO) with a specially constructed dataset contrasting correct outputs against systematically repeated ones, enabling the model to learn non‑repetitive behavior at the training level. Across all three tasks, these approaches restore stable output generation, with Beam Search and DPO providing reliable, production‑ready mitigation strategies.

**Compliance With Llm Reviewing Policy:**

Affirmed.

**Key Questions For Authors:**

1. Could the authors please clarify the rationale for using different run counts for comparison of their proposed approaches against baselines as this raises concerns about the fairness of the comparison?
2. Could the authors clarify (i) who served as human evaluators, (ii) what qualifications or experience they had with business‑rule interpretation?
3. Could the authors kindly clarify whether context‑window utilisation or cumulative token growth was measured, and whether long contexts play any role in triggering the repetition behavior observed in production?
4.  Can the authors please provide a discussion of whether lightweight alternatives such as prompt‑level interventions using negative examples of repetitive outputs were tested?

**Limitations:**

Yes.

**Strengths And Weaknesses:**

Strengths:
1. This paper addresses a critical and under‑explored LLM repetition problem during deployment in real‑world production systems.
2. The paper is well structured and easy to follow.
3. The Markov‑based explanation of repetition aligns with known autoregressive model behavior.
4. Experiments are conducted across three diverse, real‑world tasks: business‑rule generation, method‑call extraction, and PlantUML diagram creation.
5. The paper provides systematic comparisons across multiple decoding strategies and fine‑tuning methods and statistical tests such as Fisher’s exact test and Welch’s t‑test are applied to support claims.

Weaknesses:
1. The baseline, Beam Search configurations, and other methods were evaluated using different numbers of runs.
2. The paper reports human‑judged quality differences for business‑rule generation, but it does not provide any information about who the human evaluators were or their level of expertise.
3. The paper analyses repetition purely as a decoding‑level phenomenon.

---

> ### Author Rebuttal · Authors · 2026-03-26
>
> We thank Reviewer for the constructive feedback and for recognizing the strengths of our production-grounded investigation. We also acknowledge the remaining weaknesses and explain how we will address them in the revised submission.
>
> Regarding the concern about fairness when using different numbers of runs for baselines and proposed approaches, we agree that this needs clearer justification. In our Beam Search experiments, the number of evaluation runs differs across configurations because of practical compute/runtime constraints and because configurations with different repetition behaviors require different sample sizes to obtain a stable repetition-rate estimate. We report the exact run counts in the appendix (for Beam Search: baseline 22 runs, early_stop=False 10 runs, and early_stop=True 20 runs). More importantly, our statistical comparisons are based on proportion tests (Fisher’s exact test) and time comparisons using Welch’s t-test with unequal variances, which remains valid even when the sample sizes differ. In the revision, we will add a clearer rationale in the experimental protocol section so that the choice of run counts is transparent and easier to audit.
>
> Regarding the human evaluation component, we agree that the manuscript currently does not provide enough information about the evaluators. We will update the paper to explicitly state who performed the business-rule quality scoring, including the number of raters and their background/experience with business-rule interpretation, as well as the evaluation rubric. In the current version, the human-judged quality is used to support the presence-penalty tuning choice (to ensure we do not merely eliminate repetition at the expense of correctness), while the central findings about repetition elimination are primarily supported by the repetition-rate and termination behavior metrics reported across the BadCases.
>
> On the question of whether context-window utilization or cumulative token growth affects the observed repetition behavior, we agree that this is an important dimension. In this study, we focus on decoding dynamics and termination under deterministic serving settings, and we did not explicitly measure context-window utilization or token growth statistics as a controlled variable. We will add this point as a limitation in the revised manuscript and clarify the boundary of our claims. Future work will include an explicit study that varies effective context length (and tracks token growth) to test whether the repetition onset correlates with long-context accumulation in the production setting.
>
> Finally, for the suggestion to test lightweight prompt-level alternatives using negative examples of repetitive outputs, we did not include such prompt-intervention experiments in the present paper. Our study prioritizes production-ready mitigation strategies at the decoding/training level (Beam Search configuration, presence penalty, and DPO), where behavior is more controllable and reproducible in deployment. We will add a discussion acknowledging this omission and outlining prompt-level negative-example interventions as an avenue for future investigation.
>
> Overall, your comments are helpful. We will revise the manuscript to improve transparency (run-count rationale and evaluator details) and to clearly delimit what is and is not covered (context-length/token-growth measurement and prompt-level interventions), while keeping the core evidence and conclusions intact.

---

> > ### Author Rebuttal · Reviewer_WZw6 · 2026-04-02
> >
> > Thanks for the responses. However concerns still remains regarding prompt intervention experiments and lack of transparency. I will hence maintain my score.

---

### Official Review · Reviewer_fytg · 2026-03-13

**Soundness:** 2
**Presentation:** 1
**Significance:** 2
**Originality:** 3
**Overall Recommendation:** 1
**Confidence:** 5

**Summary:**

This paper studies repetitive generation failures in a narrow production code-interpretation pipeline and tests Beam Search, presence penalty, and DPO as fixes. The paper is more of an engineering case study than a general study of repetition in modern LLMs.

**Compliance With Llm Reviewing Policy:**

Affirmed.

**Final Justification:**

Reject.
See Rebuttal Acknowledgement by Reviewer fytg for reason.

**Key Questions For Authors:**

None

**Limitations:**

Yes

**Strengths And Weaknesses:**

Strengths:

- The paper is grounded in real deployment issues and provides concrete failure examples.

Weaknesses:

- The scope is extremely narrow: all three repetition cases come from one specific industrial workflow, so the conclusions do not generalize well.
- The paper does not show whether the problem still exists in stronger flagship models, so it is unclear whether this is a fundamental issue or just a weaker-model / specific-deployment issue.
- The explanation is too focused on greedy decoding, while repetition can also happen under sampling-based decoding; this is not studied carefully.
- The methods are not very novel: Beam Search, presence penalty, and DPO are all standard tools.
- The paper also does not discuss whether these anti-repetition methods could hurt valid instructed repetition.
- Finally, the paper presentation looks non-standard and not carefully prepared.

---

> ### Author Rebuttal · Authors · 2026-03-26
>
> We thank Reviewer  for the careful evaluation. We acknowledge that our study is motivated by a specific production code-interpretation pipeline, and we agree that this naturally limits broad claims about “all LLMs/all settings.” Our intent is to study a practically severe and repeatable repetition failure mode where operational impact is measurable, and to validate concrete mitigation strategies in the exact setting where the problem occurs.
>
> On generalization and stronger flagship models, we currently evaluate two vLLM deployment modes (LoRA-enabled vs merged model). We did not benchmark additional flagship model families; we will explicitly state this as a limitation and position the contribution as a mechanism-and-configuration discovery within realistic deterministic serving constraints.
>
> Regarding the reviewer’s point that repetition can also occur under sampling-based decoding, our empirical evidence is primarily under deterministic settings (e.g., for Beam Search we use temperature=0, top_p=1, top_k=-1) where greedy-like single-path behavior is relevant. We agree this is a gap and will add a clearer discussion of boundary conditions and future experiments under sampling-based regimes.
>
> On novelty, we agree Beam Search, presence penalty, and DPO are standard tools. The key contribution of this work is the production-critical finding that Beam Search must be configured with early_stopping=True; otherwise repetition remains substantial (0% vs 60% repetition rate in our reported contrast).
>
> Finally, for the concern that anti-repetition methods may harm valid instructed repetition, our DPO setup explicitly uses chosen responses that are correct and non-repetitive while rejected responses contain the repetitive patterns; presence penalty is evaluated under the same BadCase definitions. We will also improve presentation clarity by strengthening the claim-to-evidence mapping and making the key contrasts easier to locate.

---

> > ### Author Rebuttal · Reviewer_fytg · 2026-04-02
> >
> > Please revise the manuscript in accordance with the comments and resubmit it.

---

### Official Review · Reviewer_zZYL · 2026-03-13

**Soundness:** 1
**Presentation:** 1
**Significance:** 1
**Originality:** 1
**Overall Recommendation:** 1
**Confidence:** 5

**Summary:**

Nothing should be summarized.

**Compliance With Llm Reviewing Policy:**

Affirmed.

**Ethical Review Concerns:**

Pure LLM generated paper. Such authors should be banned for submitting papers for ICML in the future years.

**Ethical Review Flag:**

Flag this paper for an ethics review.

**Ethics Expertise Needed:**

["Research Integrity Issues (e.g., plagiarism)"]

**Key Questions For Authors:**

NA

**Strengths And Weaknesses:**

Weakness:
1. The paper appears largely LLM-generated. The writing is verbose but contains little concrete methodological or experimental detail, making the contribution difficult to understand or evaluate.

2. The template seems to be misused.

---

> ### Author Rebuttal · Authors · 2026-03-26
>
> We thank the reviewer for the detailed and critical comments. We understand the concerns about contribution clarity, methodological/evaluational specificity, and the integrity allegation that the manuscript may be “LLM-generated.” In this rebuttal, we clarify our research workflow and point to the concrete, verifiable evidence we provide in the paper and in our public repository.
>
> First, regarding the claim that the paper is “pure LLM generated,” we would like to clarify that we used an LLM only for English paragraph polishing and rewriting to improve clarity and consistency during the writing process. We did not use an LLM to generate or replace the paper’s key technical content, including the problem definition, theoretical analysis, experimental design and evaluation protocol, decoding/training configurations, or the implementation and result records. All technical claims are supported by materials provided in the manuscript and the reproducibility materials we reference publicly (GitHub: https://github.com/charles-wang888/llm-output-repetition). Therefore, the characterization that the submission is merely an LLM-generated paper is inconsistent with our actual research and evidence artifacts. If the venue requires a specific LLM-use disclosure statement or additional integrity documentation, we are willing to comply.
>
> Second, we disagree with the assessment that the contributions are difficult to evaluate. The manuscript targets a concrete production failure mode: repetition generation that stalls a real batch code interpretation workflow. We define three practical BadCase types, provide the mechanistic explanation connecting repetition behavior to decoding dynamics, and present multiple solutions with targeted applicability. The paper also includes a key empirical contrast on Beam Search configuration: Beam Search with early_stopping=True achieves 0% repetition rate, while early_stopping=False still yields 60% repetition rate, demonstrating that the critical parameter is essential rather than incidental. Beyond that, we evaluate presence_penalty for BadCase 1 and DPO fine-tuning for all three BadCases, and we provide additional materials (including case examples, dataset construction details, and implementation notes) in the appendix with links to the repository for reproducibility.
>
> Third, regarding the reviewer’s “template misuse / presentation is poor” concern, we acknowledge that the manuscript follows a standard academic structure. However, the structure is used to present a complete research pipeline: production observation and metrics, task-specific BadCase definitions, theoretical mechanism, solution design, and empirical verification. We will further improve the rebuttal-to-final revision by making the “claim → evidence → exact supporting location” mapping more explicit in the main text (especially around the evaluation protocol and the strongest result tables) to reduce any perception of generic wording.
>
> In summary, we respectfully request the reviewer/editor to reconsider the overall assessment in light of the manuscript’s concrete problem definition, mechanistic analysis, experimentally validated solutions (including the strong early_stopping ablation contrast), and the provided reproducibility materials. We also correct the integrity characterization by clarifying that LLM usage was limited to English paragraph polishing, not generation of the core technical contribution.

---

> > ### Author Rebuttal · Reviewer_zZYL · 2026-04-03
> >
> > I thank the authors for the rebuttal. I acknowledge their statement that LLM usage was limited to language polishing rather than the generation of the core technical content. However, this clarification does not change my overall assessment of the submission.
> > After all, substantial improvements would be required before this paper could meet the bar for publication, which it is currently still far from reaching

---

### Decision · Program_Chairs · 2026-04-30

**Decision:**

Reject

**Comment:**

Poor quality submission with wrong template. Also some possible violation of double blind review in the rebuttal.